# Tumor Response, Disease Control, and Progression-Free Survival as Surrogate Endpoints in Trials Evaluating Immune Checkpoint Inhibitors in Advanced Non-Small Cell Lung Cancer: Study- and Patient-Level Analyses

**DOI:** 10.3390/cancers15010185

**Published:** 2022-12-28

**Authors:** Nobuyuki Horita

**Affiliations:** Chemotherapy Center, Yokohama City University Hospital, Yokohama 232-0024, Japan; horitano@med.yokohama-cu.ac.jp; Tel.: +81-45-787-2800

**Keywords:** immune checkpoint inhibitors, response evaluation criteria in solid tumors, progression-free survival, endpoint determination

## Abstract

**Simple Summary:**

How tumor response and progression-free survival (PFS) reflect the overall survival (OS) in advanced non-small cell lung cancer (NSCLC) clinical trials with immune checkpoint inhibitors (ICI) have not been clarified. This study validated the uses of objective response rate and PFS for NSCLC trials with ICI through an individual-patient level and a trial level.

**Abstract:**

**Background**: To assess the usefulness of tumor response and progression-free survival (PFS) as surrogates for overall survival (OS) in non-small cell lung cancer (NSCLC) trials with immune checkpoint inhibitors (ICI), which have not been confirmed. **Methods**: Patient- and trial-level analyses were performed. The Response Evaluation Criteria in Solid Tumors was preferred for image assessment. For trial-level analysis, surrogacy was assessed using the weighted rank correlation coefficient (r) following “reciprocal duplication.” This method duplicates all plots as if the experimental and the reference arms were switched. Monte Carlo simulations were performed for evaluating this method. **Results**: A total of 3312 cases were included in the patient-level analysis. Patients without response (first line (1L): hazard ratio (HR) 1.95, 95% confidence interval (CI) 1.71–2.23; second or later line (2L-): HR 4.22, 95% CI 3.22–5.53), without disease control (1L: HR 4.34, 95% CI 3.82–4.94; 2L-: HR 3.36, 95% CI 2.96–3.81), or with progression during the first year (1L: HR 3.42, 95% CI 2.60–4.50; 2L-: HR 3.33, 95% CI 2.64–4.20), had a higher risk of death. Systematic searches identified 38 RCTs including 17,515 patients for the study-level analysis. Odds ratio in the objective response rate (*N* = 38 × 2, r = −0.87) and HR in PFS (*N* = 38 × 2, r = 0.85) showed an excellent association with HR in overall survival, while this effect was not observed in the disease control rate (*N* = 26 × 2, r = −0.03). **Conclusions**: Objective response rate and PFS are reasonable surrogates for OS in NSCLC trials with ICI.

## 1. Introduction

Primary lung cancer is one of the most frequently diagnosed and fatal malignant neoplasms in the 21st century [1,2]. Most non-small cell lung cancers (NSCLC) are classified as metastatic or locally advanced at the time of diagnosis. These high-stage incurable cases and postoperative recurrent cases are usually treated with anticancer agents, which can be grouped into cytotoxic, molecularly targeted, and immune checkpoint inhibitors (ICI). Since the second half of the last century, cytotoxic agents have been known to prolong the overall survival (OS) of patients with advanced NSCLC, [3,4,5] and the subsequent development of regimens that use molecularly targeted agents and ICIs have prolonged OS even further [6].

OS is the most preferred endpoint in clinical cancer trials [7,8,9]. OS extension is directly beneficial to the patient and can be detected without observational bias [8,10,11]. Ironically, recent advances in anticancer drugs have prolonged OS sufficiently that it is difficult to follow up with patients over enough time for OS to be measured in clinical studies. Inoperable NSCLC patients with driver mutations or high programmed death ligand 1 (PD-L1) expression may have median OS spanning years, [12,13] meaning that trials that use OS as the primary endpoint are becoming less viable [14,15,16]. The popularity of surrogate endpoints, such as the objective response rate (ORR), disease control rate (DCR), and progression-free survival (PFS) has therefore, increased [8].

Several studies have assessed how tumor response and PFS reflect the OS in advanced NSCLC clinical trials [8,17]. However, such reports have generally stated that tumor response and PFS are insufficiently correlated with OS at the trial level and therefore, cannot replace OS in NSCLC RCTs. Furthermore, these studies tended to focus on cytotoxic agents and how the Response Evaluation Criteria in Solid Tumors (RECIST) based assessment systems reflect the OS in ICI treatment have not been elucidated. Lymphocytes that are activated by immunotherapy congregate in the tumor, causing the tumor to enlarge, [18] accompanied by new metastases on images even when true cancer growth is not occurring. We are concerned whether the surrogacy of tumor response and PFS to OS in ICI trials may appear inaccurate because patients who are treated with ICI often experience pseudo-progression [19]. Despite these concerns, an increasing number of clinical studies using ICI regimens have employed tumor response and PFS as the primary endpoints.

The purpose of this study was to evaluate whether surrogate endpoints, including tumor response and PFS, can act as surrogates for OS at the trial- and patient-levels when ICIs are administered for advanced NSCLC. This is the first systematic review to evaluate the surrogacy in NSCLC trials with ICI regimen using sufficient trial- and patient-level data.

## 2. Materials and Methods

### 2.1. Study Overview

This study consisted of two parts: trial- and patient-level analyses, and was designed in compliance with the Preferred Reporting Items for Systematic Reviews and Meta-analyses (PRISMA) Statement and the PRISMA of Individual Participant Data [20]. The protocol was registered at the University hospital Medical Information Network (UMIN) website (UMIN000047001) [21].

### 2.2. Patient-Level Analysis

#### 2.2.1. Data Access

The data were provided by F. Hoffmann-La Roche, Ltd. through the intermediary, Vivli, a data distribution organization.

#### 2.2.2. Study Selection

We asked Vivli to provide data from all studies in which patients with NSCLC were treated with ICI. As this was a patient-level analysis, studies were incorporated regardless of the study phase or randomization.

#### 2.2.3. Patient Selection

Patients with metastatic, locally advanced, or recurrent NSCLC and either squamous or non-squamous pathology were selected. Specific inclusion or exclusion criteria were not set for performance status, history of prior systemic chemotherapy, level of PD-L1 expression, status of EGFR-sensitizing mutations, and ALK, ROS1, or RET fusions.

### 2.3. Treatment

Patients should be treated with ICI. Combination regimes, including dual ICI, ICI plus chemotherapy, and ICI plus molecularly targeted therapy were accepted along with single-agent ICI. Only drugs that could be considered anticancer drugs according to the guidelines were eligible for this study [3,4,5]. For example, a study in which macrolide antimicrobials were incorporated into the treatment regimen was excluded. The presence or absence of prior anti-cancer therapy was not considered. Consolidation treatment before progression of the previous systemic treatment was accepted. No cases that underwent chemoradiotherapy were included. Adjuvant ICI therapy was not allowed.

### 2.4. Outcomes

RECIST v1.1-based imaging studies were of our particular interest [22]. Confirmation was not required even for a phase II trial and independent central reviews were also not required. Modified RECIST 1.1 for immune-based therapeutics (iRECIST) was not considered.

Patients were classified into complete response (CR), partial response (PR), stable disease (SD), progressive disease (PD), and not evaluable (NE) [22]. ORR was defined by (CR + PR)/(CR + PR + SD + PD + NE) and DCR was defined by (CR + PR + SD) / (CR + PR + SD + PD + NE). Then, overall survivals were compared.

Spearman’s rank correlation coefficient (r) was assessed for PFS and OS durations. Patients who survived for one year were binomially classified into those with and without progression, preceding OS comparison.

All the analyses focused on either (i) the first-line treatment or (ii) second- or later-line treatment.

#### Statistics

Survival duration was described using Kaplan–Meier curves and compared using the log-rank test and Cox-proportional hazard model. Statistical significance was set to a *p* value of 0.05. For the statistical analysis, “cor” command and “survival” package were used.

### 2.5. Study-Level Analysis

#### 2.5.1. Study Selection

RCTs of any phase in which patients with metastatic, locally-advanced, or recurrent NSCLC patients were medically treated, and written in English were included. Conference abstracts were excluded.

#### 2.5.2. Patient Selection

See patient-level analysis.

### 2.6. Treatment

See patient-level analysis.

### 2.7. Study Search

Database searches were performed using Medline, PubMed, Cochrane CENTRAL, EMBASE, and Web of Science as of 21 February 2022 (Appendix A). Review articles were searched manually (Appendix A).

The title and abstract of potentially eligible articles were screened first, after which the full text was checked.

### 2.8. Data Extraction

Data such as author names, study design, and outcomes were extracted in a standardized manner [20].

For trials that randomized patients into three arms, two were selected based on the following criteria: (i) focusing on comparison to the primary endpoint, particularly the first gate population for the hierarchical model, (ii) producing a pair of ICI and non-ICI regimens, (iii) comparisons that were featured in the original article, and (iv) arms with larger numbers of patients.

For studies that presented outcomes for plural populations, such as whole and PD-L1 positive populations, (i) the population used for the primary endpoint and (ii) the population with a larger number of patients were selected.

RECIST version 1.1 was the preferred method for response evaluation [22]; however, similar algorithms, including modified RECIST and the WHO criteria in tumor response, were accepted.

Where survival outcomes were available at several time points, data related to the protocol-specified primary endpoint were selected.

### 2.9. Outcome

The correlations with the hazard ratio (HR) of OS (HRos) were assessed for odds ratio (OR) of ORR (ORorr), OR of DCR (ORdcr), and HR of PFS (HRpfs).

Subgroup analysis focusing on first-line treatment and second- or later-line treatment was conducted.

### 2.10. Assessment of Risk of Bias

The Cochrane risk of bias tool was used to evaluate the quality of the trials [23].

### 2.11. Statistics

The weighted Spearman’s rank correlation coefficient (r) was used to assess the surrogacy [8], with the results interpreted as no (|r| < 0.2), weak (0.2 < |r| < 0.4), moderate (0.4 < |r| < 0.6), strong (0.6 < |r| < 0.8), or excellent (0.8 < |r|) correlation. The weight of each study was given by 1/(standard error of log HRos)^2. The weighted correlation was estimated using the “corr” command in the “boot” package of software R [24]. GraphPad Prism ver 9.2.0 (GraphPad Software, San Diego CA, USA) was used to draw figures.

Examining the weighted correlation between true and surrogate endpoints is a standard approach used in the assessment of the surrogacy [8]; however, we noticed that this does not work well for ICI trials because most show favorable outcomes in the experimental arm. To address this, data from available studies were “reciprocally duplicated” before the surrogacy assessment [25]. A Monte Carlo simulation was used to evaluate the r using reciprocally duplicated data as an approximation of the r obtained by random arm determination [26].

## 3. Results

### 3.1. Patient-Level Analysis

#### 3.1.1. Patient Characteristics

Data from 3312 independent patients in three phase II and four phase III studies with ICI-containing regimens were analyzed (Appendix A) [27,28,29,30,31,32,33]. This pooled cohort included 2003 (60.5%) patients with first-line treatment and 1309 (39.5%) with second- or later-line treatment (Table 1).

#### 3.1.2. Patient-Level Surrogacy

Compared to patients with response (CR + PR), those without response (SD + PD + NE) had deteriorated OS in the first-line (HR 1.95, 95% CI 1.71–2.23, *p* < 0.001, Figure 1A) and second- or later-line (HR 4.22, 95% CI 3.22–5.53, *p* < 0.001, Figure 1B) settings. Lack of disease control (PD + NE) was associated with a high risk of death in the first-line (HR 4.34, 95% CI 3.82–4.94, *p* < 0.001, Figure 1C) and second- or later-line (HR 3.36, 95% CI 2.96–3.81, *p* < 0.001, Figure 1D) settings.

PFS and OS had high or moderate rank correlation among patients who died during follow-up after the first-line (r = 0.71, *p* < 0.001) and second- or later-line (r = 0.59, *p* < 0.001) treatment (Appendix A). Patients who survived but experienced disease progression over a 12-month period showed a higher risk of death from any cause in the first-line (*n* = 1066, HR 3.42, 95% CI 2.60–4.50, *p* < 0.001, Figure 1E) and second- or later-line (*n* = 658, HR 3.33, 95% CI 2.62–4.20, *p* < 0.001, Figure 1F) settings.

### 3.2. Study-Level Analysis

#### 3.2.1. Study Selection

Out of 823 articles that were identified through the database and manual searches, 37 research papers were selected to represent 38 RCTs, including 31 phase III trials, with 17,515 patients eligible (Appendix A). According to the inclusion criteria, five and nine studies with squamous and non-squamous histology, respectively, were included, while the other 23 studies permitted any NSCLC histology. Detailed characteristics of the studies are summarized in Table 2.

#### 3.2.2. Study-Level Surrogacy and Traditional Arm Definition

The reference and experimental arms were determined as described in the original studies (Table 2). ORorr, ORpfs, HRpfs, and HRos were extracted or calculated accordingly. Most articles defined the ICI regimen as the experimental arm and the non-ICI regimen as the reference arm (Table 2). The majority of the studies demonstrated HRos and HRpfs of < 1 and ORorr of > 1 (Figure 2A–C). This is because ICI regimens almost always improve patient outcomes compared with non-ICI regimens.

ORorr (*N* = 38, r = −0.67), ORdcr (*N* = 26, r = −0.37), and HRpfs (*N* = 38, r = 0.55) were found to be meaningfully correlated with HRos (Figure 2A–C).

#### 3.2.3. The Rationale of Reciprocal Duplication

We suspected that the correlations between ORorr and HRos, and between HRpfs and HRos were underestimated because the plots were scattered in limited areas showing favorable outcomes, e.g., right lower quadrant in the ORorr plot and left lower quadrant in the HRpfs plot (Figure 2A,C). If the reference and experimental arms had been determined randomly, half of the OR and HR values would have been reciprocals of the original value (Figure 2D–F). In this scenario, substantially improved correlations for ORorr and HRpfs with HRos would have been observed (Figure 2D–F). We therefore duplicated all plots with reciprocal HR and OR values (Figure 2G–I), and found that the graphs and the r obtained using random arm determination (Figure 2D–F) were markedly similar to those produced following reciprocal duplication (Figure 2G–I).

We hypothesized that the coefficient based on random arm selection (Figure 2D–F) is approximate to the correlation coefficient obtained following reciprocal duplication (Figure 2G–I). If so, the coefficient after reciprocal duplication (Figure 2G–I) is the better statistic because it is not affected by random error. The Monte Carlo simulations confirmed that the average coefficient obtained by repeated random arm determination converged to the coefficient obtained by reciprocal duplication when a sufficiently large number of trial data were available (Appendix A).

#### 3.2.4. Study-Level Surrogacy following Reciprocal Duplication

After reciprocal duplication, both ORorr (*N* = 38 × 2, r = −0.87) and HRpfs (*N* = 38 × 2, r = 0.85) had an excellent r with HRos (Figure 2G,I), whereas ORdcr did not correlate with HRos (*N* = 26 × 2, r = −0.03, Figure 2H).

Subgroup analysis focused on the first-line treatment and second- or later-line treatment also confirmed the surrogacy (Figure 3).

## 4. Discussion

Our analysis investigated how response evaluation and PFS were associated with OS at both the study and individual levels in NSCLC trials. We confirmed that tumor response and disease control after two cycles of treatment and 12-month PFS were key determinants of OS at the individual-patient level. On the other hand, study-level analysis confirmed that ORR and PFS are excellently correlated with OS, while DCR is not. Although post-treatment crossover has traditionally been thought to affect PFS, PFS was revealed as an excellent OS surrogate at the trial-level. ORR and DCR are simultaneously available as early phase outcomes; however, no advantage was obtained from using DCR. In fact, 12 (32%) out of the 38 studies in our study-level analysis did not present DCR data even though DCR is one of the classical RECIST outcomes. Despite the lack of a systematic review, researchers may have empirically noticed that DCR is not a reasonable study-level outcome for ICI trials. Because several novel ICIs are developed year by year and a large number of new ICI regimens for NSCLC have been evaluated in trials, [3,4,5] there is a pressing demand to validate the surrogacy of commonly used early-phase endpoints. A few reviews have systematically examined how treatment response and PFS work in non-ICI NSCLC trials. In 2015, Blumenthal et al. analyzed 14 NSCLC trials that were submitted to the US Food and Drug Administration; however, they could not establish associations between ORR and OS or between PFS and OS at the trial level [17]. Our research, which included 44 studies, showed that both ORorr (r = 0.570) and HRpfs (r = 0.496) have only moderate weighted rank correlation coefficients, suggesting ORR and PFS could not replace OS.8 Although some published studies have evaluated the surrogacy in ICI trials, few studies have confirmed surrogate end points for OS in NSCLC studies [34]. Ritchie et al. corrected phase two trials of ICI for advanced solid cancers and concluded that ORR is poorly associated with OS, with 6-month PFS recommended as an end point [35]. Another study-level analysis investigating solid cancers and anti-PD-1/PD-L1 medications by Nie et al. in 2019 concluded that no RECIST criteria-based endpoints could be a valid surrogate for OS [36]. A recent systematic review by Kok in 2021 revealed the insufficient surrogacy of ORR for OS in two-arm comparisons with any cancer [37]. In short, no previous systematic reviews have proved ORR and PFS as acceptable study-level OS surrogates in RCTs for NSCLC with ICI treatment. Our data demonstrate that ORR and PFS could replace OS and may be useful primary endpoints in ICI-based NSCLC trials. We believe that these data will encourage researchers to design new clinical trials in the future. Previous studies may have presented different conclusions because of the inclusion of solid tumors other than NSCLC, the evaluation of simple correlation (Figure 2), and the smaller number of trials.

The correlation between a proxy and a true endpoint is often selected to investigate study-level surrogacy [8,17,34,35,37]. This approach has a marked drawback in that the choice of reference and experimental arms has a large impact on the correlation coefficient (Figure 2A–F). Following the conventional rule, many studies have termed the ICI regimen the experimental arm. However, patients who are treated with the ICI regimen have better outcomes, which means that scatter plots produced using this method tend to show clusters covering small areas (Figure 2A–C). When the distribution width of the values is narrow, the correlation coefficient representing surrogacy worsens.8 This issue has also potentially existed in evaluations of the surrogacies for use with cytotoxic anticancer drugs. However, this issue was not apparent until ICI became widely used because roughly half of studies had worse clinical outcomes in the experimental arm [8,17]. Additional plotting at the point-symmetric position of the original plot frees the correlation coefficient from random errors that result from the experimental arm selection (Figure 2G–I and Appendix A).

The effect of cancer immunotherapy may be delayed or obscured because the treatment effect is mediated by immune function, unlike cytotoxic chemotherapy or molecularly targeted therapy. In addition, transient tumor growth after treatment, so-called pseudo-progression, may mimic disease progression. Pseudo-progression is an enlarged tumor nodule or a newly appeared lesion after ICI therapy. This is not actual pathological disease progression, but a seeming progression due to aggregated immune-related cells and lymphocytes. Thus, there is concern that the effect of ICI may be underestimated when ORR, DCR, or PFS are set as the primary endpoints [18]. Recently, immune-related response criteria [38] and immune-related RECIST were developed to respond to these issues [25]. Nonetheless, as we have shown in this study, RECIST-based ORR after two cycles and 12-month PFS are both excellent indicators of OS (Figure 2G,I).

Surrogacy for response evaluation and PFS in ICI trials is a major concern for any solid malignancy because the ICI regimen is widely used for various cancers. A limitation of this study is that no data on cancers other than NSCLS were provided, although a similar analysis would be expected. Regardless, the review authors could not identify the reason for this irrelevance between DCR and OR at the study-level analysis. The other limitation is that DCR and the efficacy of immunotherapy are closely dependent on initial staging and concomitant therapies; however, our study could not evaluate this issue [39].

## 5. Conclusions

In conclusion, this systematic review evaluates surrogacy in NSCLC trials with ICI regimen using sufficient trial- and patient-level data. Tumor response and PFS were appropriate predictors of OS at the patient-level. Monte Carlo simulation showed that reciprocal duplication is a reasonable strategy to apply before surrogacy evaluation at the study level. After reciprocal duplication, ORorr and HRpfs showed an excellent correlation with HRos although this relationship was not observed for ORdcr. ORR and PFS are reasonable surrogates for OS in NSCLC trials with ICI.

## Figures and Tables

**Figure 1 cancers-15-00185-f001:**
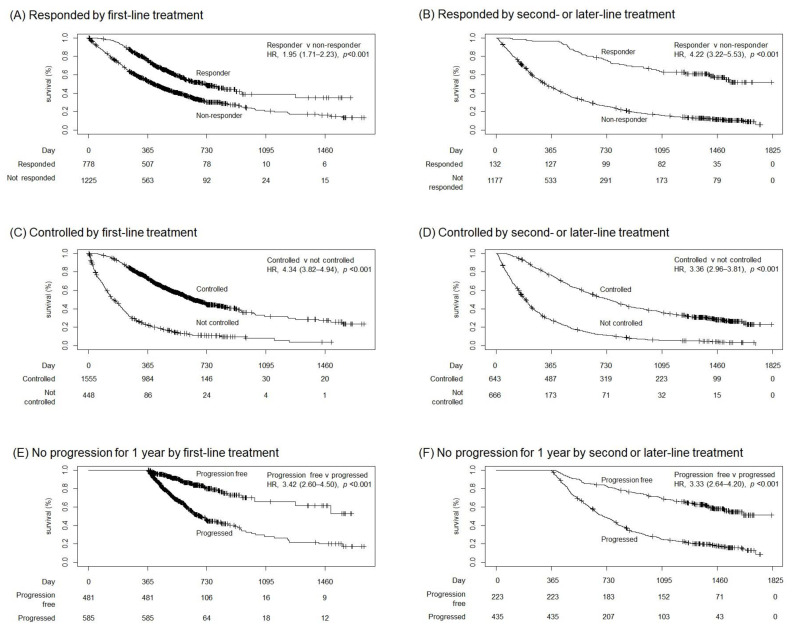
Kaplan–Meier estimates of overall survival based on response evaluation after two cycles and based on 1-year progression-free survival. Responded, complete response, and partial response; controlled, complete response, partial response, and disease control. HR, hazard ratio; brackets, 95% confidence interval.

**Figure 2 cancers-15-00185-f002:**
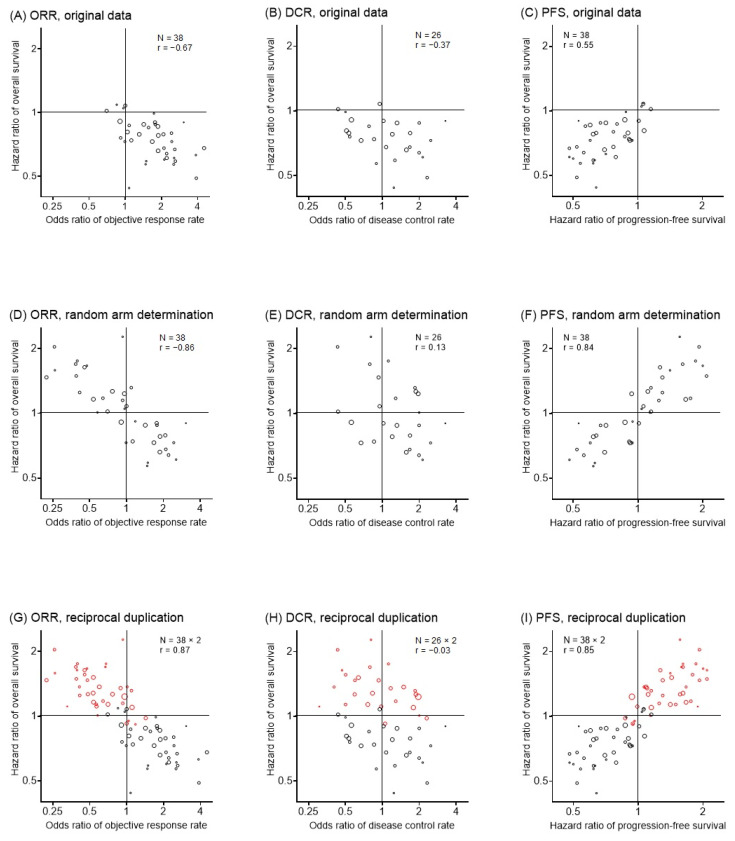
Study-level correlation. Each circle represents one randomized trial. Circle size represents study weight. Red circles indicate the point-symmetry of the original samples (black circles) to produce reciprocal duplication. *N*, number of trials; r, weighted rank correlation coefficient. ORR, objective response rate; DCR, disease control rate; PFS, progression-free survival.

**Figure 3 cancers-15-00185-f003:**
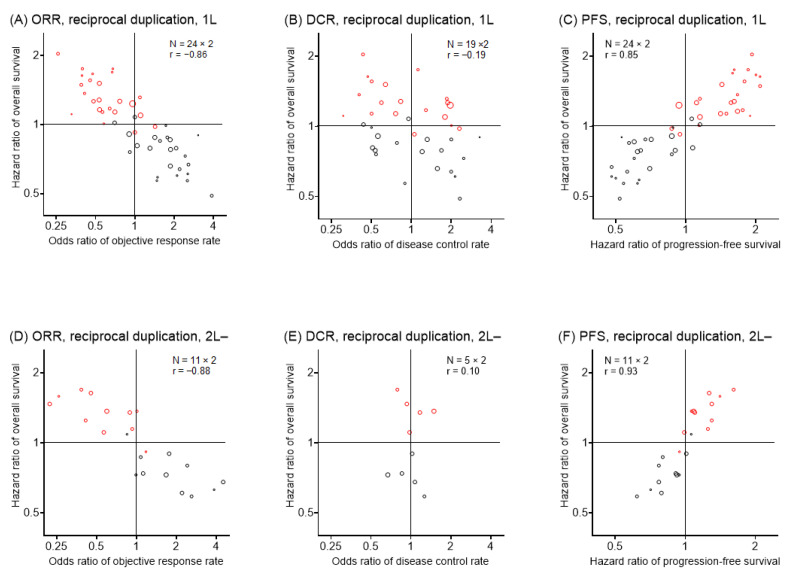
Study level-surrogacy following reciprocal duplication, subgroup analysis. Each circle represents one trial. Circle size represents study weight. Red circles indicate the point-symmetry of the original samples (black circles) for reciprocal duplication. *N*, number of trials; r, weighted rank correlation coefficient. ORR, odds ratio; DCR, disease control rate; PFS, progression-free survival. 1L, first-line subgroup; 2L-, second- or later-line subgroup.

**Table 1 cancers-15-00185-t001:** Patient characteristics for independent patient data analysis.

	1st Line	2nd or Later Line
*N*	2003	1309
Age (year)	64 (58–70)	63 (57–70)
Sex		
Men	1328 (66.3%)	803 (61.3%)
Women	675 (33.7%)	506 (38.7%)
Race		
Asian	199 (9.9%)	199 (15.2%)
Black or African American	36 (1.8%)	24 (1.8%)
White	1693 (84.5%)	1027 (78.5%)
Other/unknown	75 (3.7%)	59 (4.5%)
Pathology		
Squamous	677 (33.8%)	364 (27.8%)
Non-squamous	1309 (65.4%)	945 (72.2%)
Unknown	17 (0.8%)	0 (0.0%)
Stage		
IA	60 (3.0%)	39 (3.0%)
IB	44 (2.2%)	56 (4.3%)
IIA	46 (2.3%)	49 (3.7%)
IIB	48 (2.4%)	75 (5.7%)
IIIA	134 (6.7%)	193 (14.7%)
IIIB	96 (4.8%)	159 (12.1%)
IV (not specified for A/B)	1489 (74.3%)	55 (4.2%)
IVA	27 (1.3%)	310 (23.7%)
IVB	24 (1.2%)	346 (26.4%)
Unknown	35 (1.7%)	27 (2.1%)
TD-L1 (IC)		
0–0.9	917 (45.8%)	326 (24.9%)
1–4.9	645 (32.2%)	315 (24.1%)
5–9.9	212 (10.6%)	207 (15.8%)
10–49.9	191 (9.5%)	321 (24.5%)
50–100	38 (1.9%)	131 (10.0%)
Unknown	0 (0.0%)	9 (0.7%)
PD-LI (TC)		
0–0.9	1229 (61.4%)	550 (42.0%)
1–4.9	205 (10.2%)	292 (22.3%)
5–9.9	73 (3.6%)	61 (4.7%)
10–49.9	241 (12.0%)	181 (13.8%)
50–100	255 (12.7%)	218 (16.7%)
Unknown	0 (0.0%)	7 (0.5%)
Smoking history		
Never	290 (14.5%)	239 (18.3%)
Previous	1224 (61.1%)	901 (68.8%)
Current	489 (24.4%)	169 (12.9%)

Brackets indicate percentage or interquartile range.

**Table 2 cancers-15-00185-t002:** Characteristics of trials used in the study-level analysis.

	Country	Phase	Patho	Stage	PD-L1 Status	Driver	PS	Line	Imaging Evaluation	Treatment	ROB H/U/L
Antonia (2017) PACIFIC	USA	III	NSCLC	IIIa, IIIb	Any	ANY	0–1	After CCR	RECIST ICR	Dur (10 mg/kg) q2w No active treatment	0/0/6
Barlesi (2018) JAVELIN Lung 200	South Korea	III	NSCLC	IIIb, IV, Rec	TC > 1%	EGFR(-), ALK(-)	0–1	2–4	RECIST ICR	Avel (10 mg/kg) q2w Dtx (75 mg/m2) q3w	1/0/5
Borghaei (2015) CheckMate057	USA	III	NSQ	IIIb, IV	Any	Any	0–1	2–3	RECIST	Niv (3 mg/kg) q2w Dtx (75 mg/m2) q3w	2/0/4
Boyer (2021) KEYNOTE-598	Australia	III	NSCLC	IV	TC > 50%	EGFR(-), ALK(-)	0–1	1	RECIST, ICR	Pemb (200 mg) q3w + Ipi (1 mg/kg) q6w Pemb (200 mg) q3w	0/0/6
Brahmer (2015) CheckMate017	USA	III	SQ	IIIb, IV	Any	Any	0–1	2	RECIST,	Niv (3 mg/kg) q2w Dtx (75 mg/m2) q3w	2/0/4
Carbone (2017) CheckMate026	USA	III	NSCLC	IV, Rec	TC > 5%	EGFR(-), ALK(-)	0–1	1	RECIST, ICR	Niv (3 mg/kg) q2w Platinum doublet	1/0/5
Fehrenbacher (2016) POPLAR	USA	II	NSCLC	Adv, Met	Any	Any	0–1	2–3	RECIST	Atz (1200 mg) q3w Dtx (75 mg/m2) q3w	2/0/4
Gandhi (2018) KEYNOTE-189	USA	III	NSQ	Met	Any	EGFR(-), ALK(-)	0–1	1	RECIST, ICR	Pemb (200 mg) + Platinum + Pemt (500 mg/m2) q3w Platinum + Pemt (500 mg/m2) q3w	1/0/5
Gettinger (2021) Lung-MAP S1400I	USA	III	SQ	IV	Any	Any	0–1 (Z)	2-	RECIST	Niv (3 mg/kg) q2w + Ipi (1 mg/kg) q6w Niv (3 mg/kg) q2w	2/0/4
Govindan (2017) Study 104	USA	III	SQ	IV, Rec	Any	Any	0–1	1	mWHO	Ipi (10 mg/kg) + Cbdca (AUC 6) + Ptx (175 mg/m2) q3w Cbdca (AUC 6) + Ptx (175 mg/m2) q3w	0/0/6
Hellmann (2019) CheckMate227	USA	III	NSCLC	IV, Rec	TC >1%	EGFR(-), ALK(-)	0–1	1	RECIST ICR	Niv (3 mg/kg) q2w + Ipi (1 mg/kg) q6w Platinum doublet	1/0/5
Hensing (2021) Alliance 09	USA	II	NSCLC	IV	Any	EGFR(-), ALK(-)	0–1	no previous ICI	RECIST, ICR	Cbdca doublet q3w x4 followed by Pemb (200 mg) q3w x4 Pemb (200 mg) q3w x4 followed by Cbdca doublet q3w x4	1/0/5
Herbst (2016) KEYNOTE-010	USA	II/III	NSCLC	Adv	TC >1%	Any	0–1	2-	RECIST, ICR	Pemb (10 mg/kg) q3w Dtx (75 mg/m2) q3w	1/0/5
Herbst (2020) IMpower110	USA	III	NSCLC	IV	IC>10% or TC>50%	EGFR(-), ALK(-)	0–1	1	RECIST	Atz (1200 mg) q3w Dtx (75 mg/m2) q3w	1/0/5
Jotte (2020) IMpower131	USA	III	SQ	IV	Any	Any	0–1	1	RECIST	Atz (1200 mg) + Cbdca (AUC 6) + nPtx (100 mg/m2) q3w Cbdca (AUC 6) + nPtx (100 mg/m2) q3w	1/0/5
Jung (2022) NCT03656094	South Korea	II	NSCLC	Adv	Any	EGFR(-), ALK(-)	0–1	2–3	RECIST	Pemb (200 mg) q3w + Single-agent Chemotherpay Single-agent Chemotherpay	0/0/6
Langer (2016) KEYNOTE-021	USA	II	NSQ	III, IV	Any	EGFR(-), ALK(-)	0–1	1	RECIST	Pemb (200 mg) + Cbdca (AUC 5) + Pemt (500 mg/m2) q3w Cbdca (AUC 5) + Pemt (500 mg/m2) q3w	2/0/4
Leighl (2021) CCTG BR34	Canada	II	NSCLC	IVa, IVb	Any	EGFR(-), ALK(-)	0–1	1	RECIST	Durv (1500 mg) + Trem (75 mg) + Platinum doublet q3w Durv (1500 mg) + Trem (75 mg)	2/0/4
Lynch (2012) NCT00527735	Netherlands	II	NSCLC	IIIb, IV	Any	EGFR(-), ALK(-)	0–1	1	mWHO, ICR	Ipi (10 mg/kg) + Cbdca (AUC 6) + Ptx (175 mg/m2) q3w Cbdca (AUC 6) + Ptx (175 mg/m2) q3w	0/0/6
Mok (2019) KEYNOTE-042	HK	III	NSCLC	LocAdv, Met	TC>1%	EGFR(-), ALK(-)	0–1	1	RECIST, ICR	Pemb (200 mg) q3w Cbdca doublet	1/0/5
Nishio (2021) IMpower132	Japan	III	NSQ	IV	Any	Any	0–1	1	RECIST	Atz (1200 mg) + Platinum + Pemt (500 mg/m2) q3w Platinum + Pemt (500 mg/m2) q3w	2/0/4
Paz-Ares (2018) KEYNOTE-407	Spain	III	SQ	IV	TC >50%	Any	0–1	1	RECIST, ICR	Pemb (200 mg) +Cbdca (AUC 6) + (Ptx (200 mg/m2) or nPtx (100 mg/m2)) q3w Cbdca (AUC 6) + (Ptx (200 mg/m2) or nPtx (100 mg/m2)) q3w	0/0/6
Paz-Ares (2021) CheckMate9LA	Spain	III	NSCLC	IV, Rec	Any	EGFR(-), ALK(-)	0–1	1	RECIST, ICR	Niv (360 mg) q3w + Ipi (1 mg/kg) q6w + Platinum doublet q3w Platinum doublet q3w	1/0/5
Planchard (2020) ARCTIC-A	France	III	NSCLC	IIIb, IV	TC >25%	EGFR(-), ALK(-)	0–1	3-	RECIST	Durv (10 mg/kg) q2w Gem and Vin or Erl	2/0/4
Planchard (2020) ARCTIC-B	France	III	NSCLC	IIIb, IV	TC <25%	EGFR(-), ALK(-)	0–1	3-	RECIST	Durv (20 mg/kg) + Trem (1 mg/kg) q4w Gem and Vin or Erl	2/0/4
Reck (2016) KEYNOTE-024	USA	III	NSCLC	IV	TC>50%	EGFR(-), ALK(-)	0–1	1	RECIST, ICR	Pemb (200 mg) q3w Platinum doublet	1/0/5
Rittmeyer (2017) OAK	USA	III	NSCLC	IIIb-IV	IC>1% or TC>1%	Any	0–1	2–3	RECIST	Atz (1200 mg) Dtx (75 mg/m2) q3w	2/0/4
Rizvi (2020) MYSTIC	USA	III	NSCLC	IV	TC>25%	EGFR(-), ALK(-)	0–1	1	RECIST, ICR	Durv (20 mg/kg) q4w Platinum doublet	1/0/5
Sezer (2021) EMPOWER-Lung 1	Turkey	III	NSCLC	IIIb, IIIc, IV	TC >50%	EGFR(-), ALK(-), ROS1(-)	0–1	1	RECIST, ICR	Cemi (350 mg) q3w Platinum doublet	1/0/5
Socinski (2018) IMpower150	Germany	III	NSQ	IV, Rec	Any	EGFR(-), ALK(-)	0–1	1	RECIST, ICR	Atz (1200 mg) + Bev (15 mg/kg) + Cbdca (AUC 6) + Ptx (200 mg/m2) Bev (15 mg/kg) + Cbdca (AUC 6) + Ptx (200 mg/m2)	1/0/5
Sugawara (2021) ONO-4538-52/TASUKI-52	Japan	III	NSQ	IIIb, IV	Any	EGFR(-), ALK(-), ROS1(-)	0–1	1	RECIST, ICR	Niv (360 mg) + Cbdca (AUC 6) + Ptx (200 mg/m2) + Bev (15 mg/kg) Cbdca (AUC 6) + Ptx (200 mg/m2) + Bev (15 mg/kg)	0/0/6
West (2019) IMpower130	Italy	III	NSQ	IV	Any	EGFR(-), ALK(-)	0–1	1	RECIST, ICR	Atz (1200 mg) q3w + Cbdca (AUC 6) q3w + nPtx (100 mg/m2) q1w Cbdca (AUC 6) q3w + nPtx (100 mg/m2) q1w	1/0/5
Wu (2019) CheckMate078	China	III	NSCLC	IIIb, IV	Any	EGFR(-), ALK(-)	0–1	2	RECIST	Niv (3 mg/kg) q2w Dtx (75 mg/m2) q3w	2/0/4
Yang (2020) ORIENT-11	China	III	NSQ	IIIb, IIIc, IV	Any	EGFR(-), ALK(-)	0–1	1	RECIST	Sint (200 mg) + Platinum +Pemt (500 mg/m2) q3w Platinum + Pemt (500 mg/m2) q3w	0/0/6
ZhouC (2021) ORIENT-12	China	III	SQ	IIIb, IIIc, IV	Any	EGFR(-), ALK(-)	0–1	1	RECIST, ICR	Sint (200 mg) + Platinum +Gem (1000 mg/m2, d 1, 8) q3w Platinum +Gem (1000 mg/m2, d 1, 8) q3w	0/0/6
ZhouC (2022) GEMSTONE-302	China	III	NSCLC	IV	Any	EGFR(-), ALK(-), ROS1(-), RET(-)	0–1	1	RECIST, ICR	Suge (1200 mg) + Cbdca doublet q3w Cbdca doublet q3w	0/0/6
ZhouQ (2022) GEMSTONE-301	China	III	NSCLC	III	Any	EGFR(-), ALK(-), ROS1(-)	0–1	After CCR	RECIST, ICR	Suge (1200 mg) No active treatment	0/0/6
ZhouC (2021) CameL	China	III	NSQ	IIIb, IV	Any	EGFR(-), ALK(-)	0–1	1	RECIST, ICR	Camr (200 mg) + Cbdca (AUC 5) + Pemt (500 mg/m2) q3w Cbdca (AUC 5) + Pemt (500 mg/m2) q3w	1/0/5

Patho, pathology; NSCLC, non-small cell lung cancer; SQ, squamous cell carcinoma; NSQ, non-squamous cell carcinoma. Rec, recurrent; Adv, advanced; Met, metastasis; LocAdv, locally advanced. PD-L1, programmed death ligand 1; TC, tumor cells; IC, tumor-infiltrating immune cells; >1%, 1% or higher; >5%, 5% or higher; >10%, 10% or higher; >50%, 50% or higher. PS, performance status (if not specified, Eastern Cooperative Oncology Group PS); Z, Zubrod. After CCR, consolidation therapy after combined chemoradiotherapy. RECIST, Response Evaluation Criteria in Solid Tumors version 1.1; ICR, independent central review. Niv, nivolumab; Pemb, pembrolizumab; Sint, sintilimab; Cemi, cemiplimab; Camr, camrelizumab; Avel, avelumab; ATZ, atezolizumab; Durv, durvalumab; Suge, sugemalimab; Ipi, ipilimumab; Trem, tremelimumab. Dtx, docetaxel; Pemt, pemetrexed; Cbdca, carboplatin; nPtx, nab-paclitaxel; Ptx, paclitaxel; AUC, area under curve; Gem, gemcitabine; Vin, vinorelbine; Erl, erlotinib; Bev, bevacizumab. q3w, every 3 weeks. ROB, Cochrane risk of bias; H/U/L, high/unclear/low risk of bias.

## Data Availability

Restrictions apply to the availability of these data. Data was obtained from Hoffmann-La Roche and are available Vivli, Inc. (https://vivli.org/) with the permission of Vivli, Inc.

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
