# Peer review of "Tumor Response, Disease Control, and Progression-Free Survival as Surrogate Endpoints in Trials Evaluating Immune Checkpoint Inhibitors in Advanced Non-Small Cell Lung Cancer: Study- and Patient-Level Analyses"

_cancers, 2022, doi:10.3390/cancers15010185_

Round 1

Reviewer 1 Report

I am writing about the paper " Tumor response, disease control, and progression-free survival 2 as surrogate endpoints in trials evaluating immune checkpoint 3 inhibitors in advanced non-small cell lung cancer: study- and 4 patient-level analyses". Here they discussed how the treatment of immune checkpoint inhibitors in NSCLC improves the tumor burden and progression-free survival,
which increases the overall survival of the patient. This topic is interesting. I have no big issues.

1. Sample size is enough to justify these findings.

2. Overall the data presentation is good.

Author Response

We are grateful for your deliberate responses to our manuscript.

Reviewer 2 Report

The author evaluated whether surrogate endpoints namely objective response rate (ORR), disease control rate (DCR), progression-free survival (PFS) can surrogate overall survival (OS) at independent-patient- and trial-level when immune checkpoint inhibitor (ICI) is administered for advanced non-small cell lung cancer (NSCLC). The analysis showed that ORR and PFS were appropriate predictor of OS at the patient-level, and odds ratio of ORR and hazard ratio of PFS were associated with hazard ratio of OS. ORR and PFS are excellently considered to be reasonable surrogates for OS in NSCLC trials with ICI, thus ORR and PFS can replace OS and may be useful a primary endpoint in ICI-based NSCLC trials. The validity of ORR or PFS as a surrogate for OS with ICI is uncertain and may differ by tumor type. Thus this meta-analysis would be helpful for readers. In general, this manuscript is logical and interesting, and discussed a hot topic in NSCLC treatment. However, the following points should be addressed.

While both ORR and DCR were associated with OS at the patient-level, DCR (OR) was not correlated with OS at the study level. The author should discuss why the results were different to avoid confusion for the readers. Did the lack of DCR data in the corrected studies influence the result?

As described in the Discussion section, several studies have reported that ORR or PFS could not replace OS in NSCLC patients with ICI treatment (line 284-297). Please describe clearly the difference of statistical analysis methods to lead the conclusion in this paper.

Minor points

Please describe clearly the definition of ORR and DCR according to RECIST for readers (in addition to ”responder” or ”controlled” in line 102-106).

There are some typos; Figure 3. Study revel, line 289 OS.8, line 309 worsens.8

Author Response

<<Reviewer 2>>

The author evaluated whether surrogate endpoints namely objective response rate (ORR), disease control rate (DCR), progression-free survival (PFS) can surrogate overall survival (OS) at independent-patient- and trial-level when immune checkpoint inhibitor (ICI) is administered for advanced non-small cell lung cancer (NSCLC). The analysis showed that ORR and PFS were appropriate predictor of OS at the patient-level, and odds ratio of ORR and hazard ratio of PFS were associated with hazard ratio of OS. ORR and PFS are excellently considered to be reasonable surrogates for OS in NSCLC trials with ICI, thus ORR and PFS can replace OS and may be useful a primary endpoint in ICI-based NSCLC trials. The validity of ORR or PFS as a surrogate for OS with ICI is uncertain and may differ by tumor type. Thus this meta-analysis would be helpful for readers. In general, this manuscript is logical and interesting, and discussed a hot topic in NSCLC treatment. However, the following points should be addressed.

We are grateful for your deliberate responses to our manuscript.

1)

While both ORR and DCR were associated with OS at the patient-level, DCR (OR) was not correlated with OS at the study level. The author should discuss why the results were different to avoid confusion for the readers. Did the lack of DCR data in the corrected studies influence the result?

As the Reviewer 2 commented, readers may be interested in why DCR did not correlate with OS at the study level. Given the nearly null correlation between DCR and OR (Figure 2 H), absence of the correlation could not be explained by the small number of the studies. The review authors cannot identify the reason for this irrelevance. This issue was added to the limitation section.

2)

As described in the Discussion section, several studies have reported that ORR or PFS could not replace OS in NSCLC patients with ICI treatment (line 284-297). Please describe clearly the difference of statistical analysis methods to lead the conclusion in this paper.

Based on this suggestion, we additionally explain the discrepancy between previous studies and our study as below.

“Previous studies might have presented different conclusion because of the inclusion of solid tumors other than NSCLC, evaluation of simple correlation (Figure 2 A-C), and smaller number of trials.”

3) Minor points

Please describe clearly the definition of ORR and DCR according to RECIST for readers (in addition to ”responder” or ”controlled” in line 102-106).

Following your advice, the sentence was amended as the follows.

“ORR was defined by (CR + PR) / (CR+PR+SD+PD+NE) and DCR was defined by (CR + PR + SD) / (CR+PR+SD+PD+NE).”

Thank you.

4)

There are some typos; Figure 3. Study revel, line 289 OS.8, line 309 worsens.8

We appreciate the careful review by Reviewer 2. These typos were corrected.

Reviewer 3 Report

please clarify if iRECIST criteria were considered

additional data concerning smoking exposure , comorbidities and concomitant medications should be provided. Please explain better the causes of pseudoprogression to ICIs.

What about disease control rate and disease free survival in immunotherapy adjuvant setting? please clarify

since disease control rate and the efficacy of immunotheapy are closely dependent on initial staging and  concomitant therapies please include the following reference

-Crit Rev Oncol Hematol. 2019 Oct;142:26-34.

Author Response

<<Reviewer 3>>

We are grateful for your deliberate responses to our manuscript.

1)

please clarify if iRECIST criteria were considered

We additionally wrote “Modified RECIST 1.1 for immune-based therapeutics (iRECIST) was not considered.” in the method section. Even though we recognize the meaning of iRECIST, it has been less frequently used in lung cancer trials than regular RECIST.

Thank you.

2)

additional data concerning smoking exposure , comorbidities and concomitant medications should be provided.

Based on this comment, we added the data concerning smoking exposure on Table 1. Unfortunately, data for comorbidities were not available. Concomitant medications were summarized in Supplementary Table 2.

3)

Please explain better the causes of pseudoprogression to ICIs.

Another explanation for the cause of pseudoprogression was added to the discussion section. Thank you.

4)

What about disease control rate and disease free survival in immunotherapy adjuvant setting? please clarify

We did not include adjuvant therapy for this study. The following sentence was added to the method section. “Adjuvant ICI therapy was not allowed.”

5)

since disease control rate and the efficacy of immunotheapy are closely dependent on initial staging and  concomitant therapies please include the following reference

-Crit Rev Oncol Hematol. 2019 Oct;142:26-34.

We agree with the Reviewer 3 that DCR and the efficacy of immunotheapy are closely dependent on initial staging and concomitant therapies. However, our study could not evaluate this issue. This was added as one of the limitations and referred the suggested reference. Thank you.

Round 2

Reviewer 2 Report

Thank you for your response. The manuscript has been improved and is in a nice condition now.